



# Dynamic environment but no temperature change since the late Paleogene at Lühe Basin (Yunnan, China)

Caitlyn R. Witkowski[1,2], Vittoria Lauretano[1,2], Alex Farnsworth[3], Shu-Feng Li[4], Shi-Hu Li[5,6], Jan Peter Mayser[1,2], B. David A. Naafs[1,2], Robert A. Spicer[4,5,7], Tao Su[4], He Tang[4,8], Zhe-Kun Zhou[4], Paul J. Valdes[3], Richard D. Pancost[1,2]

[1]School of Earth Sciences, and Cabot Institute, University of Bristol, Bristol, BS8 1TS, UK
[2]Organic Geochemistry Unit, School of Chemistry, University of Bristol, Bristol, BS8 1QU, UK
[3]School of Geographical Sciences and Cabot Institute, University of Bristol, Bristol, BS8 1SS, UK
[4]CAS Key Laboratory of Tropical Forest Ecology, Xishuangbanna Tropical Botanical Garden, Chinese Academy of Sciences, Mengla 666303, China
[5]State Key Laboratory of Lithospheric Evolution, Institute of Geology and Geophysics, Chinese Academy of Sciences, Beijing 100029, China
[6]Lancaster Environment Centre, Lancaster University, Lancaster, LA1 4YQ, UK
[7]School of Environment, Earth and Ecosystem Sciences, The Open University, Walton Hall, Milton Keynes, MK7 6AA, UK
[8]State Key Laboratory of Isotope Geochemistry, Guangzhou Institute of Geochemistry, Chinese Academy of Sciences, Guangzhou 510640, China

*Correspondence to*: Caitlyn R. Witkowski (caitlyn.witkowski@bristol.ac.uk)

**Abstract.** The complex tectonic evolution in the Tibetan region has impacted climate, the Asian monsoon system, and the development of major biodiversity hotspots, especially since the onset of the India-Eurasia continental collision during the early Paleogene. Untangling the links between the geologic, climatic, and ecological history of the broader region can provide insights into these Earth system mechanisms, relevant for the future of our rapidly changing planet. To better understand environmental conditions across this critical time and place, we reconstruct the climatic and environmental history from a key sedimentary repository within the Lühe Basin, Yunnan, China, uniquely located between high elevation Tibet and low elevation coastal China. We investigate a 340-m long section using a multi-proxy organic geochemistry approach, complemented with sedimentological interpretations and climate model simulations. The complementary organic geochemical proxies, including *n*-alkanes, terpenoids, and hopanes, suggest that these thermally immature sediments were deposited in a dynamic environment that fluctuated between low energy floodplains and high energy fluvial systems. Our branched glycerol diakyl glycerol tetraether-based proxies indicate terrestrial temperatures of around 17°C ± 3 SD and our model-based temperatures indicate terrestrial temperatures for Chattian of around 19°C, consistent with the literature palaeobotany-based temperatures from the nearby Lühe town section. These combined palaeotemperatures match present-day values, suggesting that this area has not undergone significant temperature change since the early Oligocene.



## 1 Introduction

The 'Tibetan region' (i.e., Himalaya–Tibet–Hengduan mountain area) is a major feature of our planet. Since the India-
Eurasia continental collision during the early Paleogene, the Tibetan region has had a powerful and lasting impact on climate
(e.g., Raymo and Ruddiman, 1992; France-Lanord and Derry, 1997; Farnsworth et al., 2019a), the Asian monsoon systems
(e.g., Huber and Goldner, 2012), and Asian biodiversity (e.g., Li et al., 2021). Today, the Tibetan region provides the
headwaters for the ten largest rivers in Asia and consequently freshwater to nearly one-fourth of the human population. It is
also home to some of the richest biodiversity on Earth. Understanding the links between topography, climate, hydrology, and
ecology in the broader region are thus key to the history of the region (Spicer et al., 2020, and references therein), and
fundamental for managing future natural resources, including biodiversity.

However, the history of the Tibetan region remains unresolved. Following the India-Eurasia continental collision
during the early Paleogene, globally compiled marine records document a congruent decline in carbon dioxide
concentrations, growth of the Antarctic icesheet, and reorganisation of the global climate system during the late Paleogene,
and mostly notably during the Eocene Oligocene Transition (EOT; e.g., Westerhold et al., 2020). However, the few existing
terrestrial records demonstrate high heterogeneity in global change during this time, with some records suggesting no change
over this period (Retallack, 2007; Sheldon et al., 2012; Kohn et al., 2015) while others suggest cooling (e.g., Zanazzi et al.,
2007; Lauretano et al., 2021). Terrestrial records for the Tibetan region likewise show complex and heterogeneous changes
in biodiversity (e.g., Li et al., 2021), which had downstream impacts on e.g., Yunnan, one of Asia's biodiversity hotspots
situated in southwestern China along the SE Tibetan margin (Li et al., 2020; Spicer et al., 2020). However, apart from these
specific examples, the lack of other (well-dated) sections has hindered attempts to correlate these interior locations to the
global Cenozoic climate trends extrapolated from marine records. Reconstructing the climatic history of sedimentary basins
along the margin of Tibet is crucial to understand the connection between topographic relief and climate, their influence on
the Asian monsoon system, and their link to global climate.

Here, we reconstruct the climatic and environmental conditions during the late Paleogene in Yunnan, specifically
from a key sedimentary repository located along the SE margin of the Tibetan Plateau known as Lühe Basin. We use organic
geochemistry, sedimentology, and modelling, contextualised with palaeobotany, to develop a more robust and holistic
understanding of this region. Although there have been recent efforts to better constrain late Paleogene climatic and
environmental conditions throughout the Tibetan region using modelling and palaeobotanical tools (e.g., Su et al., 2019a;
2020), few studies have used quantitative organic geochemical proxies. Because organic geochemical proxies can be
measured continuously throughout a section, they more robustly demonstrate change over time as compared with the
geologic-stage time slices used by models or with the individual and sporadic mega-fossil horizons used in palaeobotany;
thus, the addition of organic geochemistry offers new perspectives at this location. First, we tease out the sources for organic
matter and track changes in vegetation using *n*-alkanes derived from plant leaf waxes and/or mixed sources (Eglinton and
Hamilton, 1967); we also track changes in vegetation via diterpenoids and triterpenoids that are respectively diagnostic for





gymnosperms (i.e., plants with unenclosed seeds, such as conifers) and angiosperms (i.e., flowering plants with enclosed seeds) (Otto and Wilde, 2001). Next, we provide the first in-depth description and interpretation of the Lühe Basin sediment log, and we then contextualise the depositional environment through the combined analyses of the organic geochemistry and sedimentology. Then, we reconstruct mean annual temperatures across the Lühe Basin using branched glycerol dialkyl

glycerol tetraethers (brGDGTs), membrane-spanning lipids likely synthesized by bacteria and widely used as paleothermometers (Sinninghe Damsté et al., 2000; Weijers et al., 2007), and we run climate model simulations to further understand the likely temperatures for this region under different conditions. We then contextualise our organic geochemistry-based and model-based temperatures with the palaeobotany-based temperatures from a nearby site, and finally describe our findings in the context of the literature during this time period. Together, this study explores vegetation,

depositional environment, and climate results to assess the regional impact of secular climate change through the late Paleogene at this critical location.

## 2 Materials and Methods

### 2.1 Geological context

The Lühe Basin is located in Nanhua County along the southern side of the Chuxiong fault, situated in central Yunnan

Province, southwestern China (25.1416°N, 101.3738°E; Fig. 1). The Lühe Basin has been studied in detail across two outcrop sections located 2.6 km from one another: the Lühe town section and the Lühe coalmine section (the latter section is studied here; Fig. 1, Fig. 2). Both sections were initially assigned to the late Miocene based on palynological and floral evidence, as well as regional stratigraphic correlations (Zhang et al., 2007; Xu et al., 2008). However, U-Pb zircons from volcanic ashes in the Lühe town section indicated an age between 33 ± 1 Ma and 32 ± 1 Ma (Linnemann et al., 2018),

backdating at least part of the Lühe town section to the earliest Oligocene. Li et al. (2020) further constrained this age by providing a new magneto- and radio-isotopic framework for the sedimentary succession exposed in the Lühe coalmine section, the section studied here. At the Lühe coalmine section, new $^{40}Ar/^{39}Ar$ dating of feldspars within volcanic ashes exposed at 58 m in the section provides ages of 33.32 ± 0.36 Ma (sample lvb11) and 34.38 ± 0.74 Ma (sample lv5 8.0). This has been used to constrain the magnetostratigraphic interpretation of the Lühe coalmine section (Fig. 2), suggesting that the

section spans magnetochrons C15n to C9n (ca. 35-27 Ma, Gradstein, 2012, updated for Speijer et al., 2020). This matches the geologic timescale updates conducted by Xu et al. (2023), who resampled the lower 180 m of the coalmine section of Lühe Basin to provide a robust high-resolution Sr, Rb, Rb/Sr, and Ti data and cyclostratigraphy interpretations. This yields an average sedimentation rate of ca. 48 cm/kyr, consistent with the rates in other basins around the Tibetan Plateau (Li et al., 2020). Uncertainty in the $^{40}Ar/^{39}Ar$ dates from which magnetochrons were identified, however, means that those

identifications are tentative; in other words, it is possible that the studied section is slightly younger. The uncertainties in the age are important to note, as it means that the Lühe Basin may or may not cover the Eocene-Oligocene transition (EOT) that occurred at 33.9 Ma. Regardless, this timing means that the Lühe coalmine section is similar in age to the Lühe town section



(Linnemann et al., 2018), where the town section likely corresponds to between 70 to 130 m of our coalmine section. We note the uncertainties in the precise age in our discussion.

The Lühe coalmine section succession comprises alternations of organic-rich marls, mudstones, sandstones, and lignite (i.e., immature fossilised peat) deposits (Fig. 1;  Fig. 2; Table S3; Fig. S5). The lower part of the section (0-73 m) comprises small grain sizes, ranging from clay to silt, and the upper part of the section (73-340 m) generally comprises larger grain sizes from sands to gravels, interspersed with some brief intervals of organic-rich silts. A thick coal interval (ca. 4 m) at ca. 50 m from the base of the coalmine contains 11 volcanic ash layers, some of which were used for the dating described

above. The measured ca. 340-m thick profile was logged in 2018 along the SE margin of the exposed Lühe coalmine and is stratigraphically correlated with that of Li et al. (2020) and updated for Speijer et al., (2020) (Fig. 2). The sedimentology is fully described in the Supplementary Information (Table S3; Fig. S5) and expanded upon in the discussion on the depositional environment (Sect. 3.3).

## 2.2 Organic geochemistry

### 2.2.1 Sample preparation

A total of 56 samples were analysed for their biomarker content to determine the thermal maturity, depositional environment, palaeovegetation, and paleotemperatures. Samples were extracted using a microwave-assisted extraction system with dichloromethane (DCM) and methanol (MeOH) (9:1 v/v). The resulting total lipid extract was eluted with alumina column chromatography into an apolar fraction using hexane:DCM (9:1 v/v) and a polar fraction using DCM:MeOH

(1:2 v/v). Apolar fractions were analysed via GC-MS (representative examples shown in Fig. 3), and polar fractions were analysed via HPLC-MS. More detailed methodology can be found in Supplementary Material.

### 2.2.2 Indices for thermal maturity

To assess the degree of thermal maturity of the organic matter preserved in the sediments (Fig. 4), given that high thermal maturity can affect the fidelity of GDGT-based temperature reconstructions (Schouten et al., 2004; Schouten et al., 2013),

we use $n$-alkanes derived from plant leaf waxes (high-molecular-weight components) and/or mixed sources (low-molecular-weight components) and hopanes derived from bacteria. We calculated the carbon preference index (CPI), which measures the odd-over-even preference of mid- and long-chain $n$-alkanes (Bray and Evans, 1961). Odd-carbon-number $n$-alkanes are preferentially biosynthesised by higher plants (Eglinton and Hamilton, 1967), meaning terrigenous distributions have high CPIs; these decrease with degradation and thermal maturity. Here, CPI was calculated as ($\Sigma$odd ($C_{21}$"-"$C_{33}$) + $\Sigma$odd ($C_{23}$"-

"$C_{35}$)) / ($2 \times \Sigma$even ($C_{22}$"-"$C_{34}$)) to avoid overestimation of the odd-over-even preference (Marzi et al., 1993). We also calculated the stereochemistry ($\alpha$ or $\beta$) of the $C_{29}$ and $C_{31}$ hopanes at the C–17 and C–21 positions, expressed as the $\beta\beta$ / ($\beta\beta$ + $\alpha\beta$ + $\beta\alpha$) ratio, which decreases with increasing thermal maturity (e.g., Mackenzie et al., 1980).





### 2.2.3 Indices for vegetation and environmental reconstructions

The apolar fraction contained eukaryote-derived compounds (i.e., *n*-alkanes, diterpenoids, triterpenoids) that were used to
identify vegetation and environmental conditions (Figs. 2 and 4). The average chain length (ACL) of *n*-alkanes can be
indicative of the dominant source vegetation and was calculated as ACL = $\Sigma(C_n \times n) / \Sigma(C_n)$ (Eglinton and Hamilton, 1967),
based on odd *n*-alkane chain-lengths from $C_{25}$ through $C_{35}$. Because $C_{25}$ (and $C_{23}$) *n*-alkanes are produced in high abundance
by *Sphagnum* mosses and some submerged vascular macrophytes and are much less abundant in higher plants (Ficken et al.,
2000; Bush and McInerney, 2013), we excluded the $C_{25}$ *n*-alkane from our ACL; here we instead calculate ACL from odd *n*-
alkane chain-lengths from $C_{27}$ through $C_{35}$. The exclusion of $C_{25}$ helps avoid skewing the overall values so that we may more
clearly diagnose variations in vegetation e.g., grasses, angiosperms, and gymnosperms. The $C_{23}$ and $C_{25}$ *n*-alkanes are
accounted for in other indices: in the *P*-aqueous ratio ($P_{aq}$) calculated as $P_{aq} = (C_{23}+C_{25})/(C_{23}+C_{25}+C_{29}+C_{31})$ (Ficken et al.,
2000) and the $C_{23}/(C_{23}+C_{31})$ index (Nott et al., 2000), which were both used to indicate wetland conditions, given that the
likely sources for $C_{23}$ and $C_{25}$ *n*-alkanes are *Sphagnum* mosses and some submerged vascular macrophytes. CPI, as described
in 2.2.2, was used as supplementary information for interpreting changes in terrestrial input. We also used the presence and
abundance of terpenoids, compounds generally associated with specific vegetation types (Alpin and Cambie, 1964; Otto and
Wilde, 2001), in which tricyclic diterpenoids are indicative of gymnosperms (notably conifers) and nonsteroid pentacyclic
triterpenoids are indicative of angiosperms.

### 2.2.4 brGDGT indices for MAAT and pH

The polar fractions contained branched glycerol dialkyl glycerol tetraethers (brGDGTs; structures shown in Fig. S1),
membrane-spanning bacterial lipid biomarkers (Sinninghe Damsté et al., 2000) used to reconstruct temperature and pH
(Weijers et al., 2007). The polar fractions contained no (or only traces of) archaeal isoprenoidal GDGTs; these lipids are not
used for this study.

    To quantify temperature, we applied the most updated and complete global calibrations from brGDGTs relevant for
this terrestrial section, which we outline below. Briefly, the GDGTs in our section could have been produced in mineral
soils, peats, or a shallow lake environment based on our environmental reconstructions (Sect. 3.4); we therefore used soil-
calibrated mean annual air temperature (MAAT_soil, Naafs et al., 2017a), peat-calibrated MAAT (MAAT_peat; Naafs et al.,
2017b), and lake-calibrated MAAT for months above freezing (MAF_lake, Martínez-Sosa et al., 2021).

    The brGDGT-based temperature proxy is based on the correlation of MAAT with the degree of methylation of
branched tetraethers (MBT) (Weijers et al., 2007). The MBT parameter was subsequently refined by the identification and
separation of 5-methyl and 6-methyl brGDGTs (De Jonge et al., 2014), yielding MBT'$_{5me}$ based on the temperature-
dependence of 5-methyl brGDGTs alone (Fig. S1):

$$\text{MBT'}_{5me} = (Ia + Ib + Ic)/ (Ia + Ib + Ic + IIa + IIb + IIc + IIIa) \tag{1}$$




Further revision of the available global soil brGDGT data excludes from the compilation 6-methyl dominated brGDGTs
from arid and/or alkaline soils (Naafs et al., 2017a), leading to

$$MAAT_{soil} = 40.01 \times MBT'_{5me} - 15.25 \quad (n=350, R^2=0.60, RMSE=5.3°C). \quad (2)$$

The degree of cyclization of branched tetraethers (CBT) correlates with pH in mineral soils (Weijers et al., 2007). The CBT
index was later revised into CBT' (De Jonge et al., 2014), which included 6-methyl brGDGTs and improved the correlation
with pH:

$$CBT' = {}^{10}log\,[(Ic+IIa'+IIb'+IIc'+IIIa'+IIIb'+IIIc')/(Ia+IIa+IIIa)] \quad (3)$$
$$pH = 7.15 + 1.59 \times CBT', \quad (n=221, R^2=0.85, RSME=0.52) \quad (4)$$

Most work on brGDGTs is based on mineral soils, but brGDGTs are particularly abundant in peat deposits (Sinninghe
Damsté et al., 2000; Naafs et al., 2019). The relationship between environmental parameters and the distribution of
brGDGTs in peats led to the first peat-specific temperature and pH calibrations based on a global peat database (Naafs et al.,
2017b). The relationship between MBT'$_{5me}$ and MAAT in this case is expressed as:

$$MAAT_{peat}\,(°C) = 52.18 \times MBT'_{5me} - 23.05 \quad (n=96, R^2=0.76, RMSE=4.7°C) \quad (5)$$

while the correlation between brGDGTs and pH is defined as:

$$CBT_{peat} = log\,[(Ib+Iia'+Iib+Iib'+IIIa')/(Ia+Iia+IIIa)] \quad (6)$$
$$pH = 8.07 + 2.49 \times CBT_{peat}, \quad (n=51, R^2=0.85, RSME=0.8) \quad (7)$$

brGDGTs have also been found in freshwater aquatic environments, such as lakes (e.g., Tierney and Russell, 2009; Raberg
et al., 2021) and rivers (e.g., Zell et al., 2013). Here, we apply the latest and most extensive global calibration for lacustrine
settings (Martínez-Sosa et al., 2021) based on the MBT'$_{5me}$ index which correlates with mean annual temperature and
correlates most strongly with months above freezing (MAF) as:

$$MAF_{lake} = [MBT'_{5me} - 0.075]/0.030 \quad (8)$$

**2.2.5 Climate model simulations**

We utilised a suite of climate model simulations to assess the impact on Asian climate of decreasing atmospheric
concentrations of carbon dioxide ($pCO_2$) and the formation of a Southern Hemisphere icesheet using late Eocene (Priabonian
stage) and Oligocene (Rupelian and Chattian stages) boundary conditions. We employed HadCM3BL-M2.1aD (Valdes et
al., 2017), a primary model of the IPCC AR3 to AR5 experiments with a fully coupled ocean-atmosphere and dynamic
vegetation General Circulation Model (GCM) with a 3.75 x 2.5 latitude by longitude spatial grid (ca. 300 km), nineteen
vertical levels in the atmosphere and twenty vertical levels in the ocean. This version of HadCM3 has shown spatio-temporal
skill in reproducing the modern observed Asian monsoon and paleo-monsoon (Farnsworth et al., 2019a), providing
confidence in its thermodynamic and hydrologic response to perturbed forcing for the current region of interest.

Model boundary conditions (topography, bathymetry, and ice sheet configurations; at 0.5 x 0.5° resolution and
downscaled to model resolution) for each geologic stage, Priabonian (ca. 36 Ma), Rupelian (ca. 31 Ma), and Messinian (ca.
25 Ma), are provided by Getech Plc. Stage-specific solar luminosity was calculated using the methods of Gough (1981).



$p$CO$_2$ values were prescribed at 1120 ppm for the Priabonian and 560 ppm for the Rupelian and Chattian, consistent with the Phanerozoic $p$CO$_2$ compilations (Foster et al., 2017; Witkowski et al., 2018). Topographic changes were also considered across the Priabonian to Rupelian based on hypotheses posed by Spicer et al., 2020 (and references therein), either as a) a

constant valley at 2.5 km elevation, b) a constant plateau at 4.5 km elevation, or c) a change from a valley at 2.5 km to a plateau at 4.5 km.

Each experiment was run for 12,422 model years to allow the surface and deep ocean to reach equilibrium and to achieve a state with no net energy imbalance at the top of the atmosphere. This spin-up is fundamental as ocean circulation can take many thousands of model years to establish its equilibrium state, which consequently has a considerable influence

on the climate signal, and thus can lead to a potentially erroneous state if not adequately spun-up (Farnsworth et al., 2019b). Climate means were calculated from the last 100-years of each simulation. Time-varying latitude and longitude plate paleo-rotations are provided for the Lühe Basin for each stage to allow for accurate comparison within the model. The paleo-coordinates (21.1°N) for Lühe were calculated using the Getech plate model.

## 3 Results and Discussion

### 3.1 Thermal maturity of sediments

The apolar fractions were used to estimate thermal maturity at the Lühe Basin coalmine section (Figs. 2, 3). The CPI ranges from 2.0 to 11.3 with a mean of $5.6 \pm 1.9$ σ (Fig. 4d). Given that thermal maturity drives CPI towards values of 1 (Bray and Evans, 1961), these samples are relatively immature. Furthermore, the sedimentology with soft clay- to siltstones and the presence of polar (e.g., brGDGTs) lipids indicate that these are indeed low maturity sediments.

We also determined the hopane isomerisation ratio of ββ/(ββ + αβ + βα), which decreases with increasing thermal maturity, such that values < 0.5 are generally thought to be associated with less reliable GDGT-based proxy estimates (Schouten et al., 2004; 2013). Here, the C$_{31}$ hopane isomerisation ratio ranges from 0.1 to 0.7 with a mean of $0.4 \pm 0.2$ σ, while the C$_{29}$ hopane isomerisation ratio ranges from 0.2 to 0.9 with a mean of $0.5 \pm 0.2$ σ (Fig. 4c). Values are slightly lower in the bottom ca. 30 m of the section. Although variable, most values are > 0.3 and there is no consistent trend

throughout the section. The large variation we observe at Lühe cannot be caused by changes in thermal maturity across such stratigraphically short intervals, and the lower values are inconsistent with the other biomarker data which clearly indicate low thermal maturity. In the context of the site, therefore, the lower hopane ratios instead indicate either allochthonous inputs from reworked kerogen as seen in marine sections (e.g., Hefter et al., 2017; Lyons et al., 2019) or accelerated isomerisation as seen in acidic peatlands (Inglis et al., 2018). The lowest hopane isomerisation ratios occur in the samples

with the highest TOC and lowest pH (Fig. S2; Fig. 4a, 3c), consistent with the latter explanation; Inglis et al. (2018) showed that the αβ-isomerisation can occur almost instantaneously in peatlands, appearing to be strongly regulated by the acidic environment. Therefore, our samples with TOC > 30% (Fig. 4a) and low pH (as low as 2.97, Fig. 4b; described in Sect. 3.4) likely represent peat-forming depositional environments that resulted in the low hopane isomerisation ratios.



In summary, these are immature sediments, giving confidence in our GDGT-based paleotemperature
reconstructions.

## 3.2 Vegetation reconstructions

Throughout the section, the *n*-alkane distribution has an odd-over-even preference (Fig. 3, Fig. 4d; Table S1), suggesting an
origin from leaf waxes and a significant input of terrestrial plants (Eglinton and Hamilton, 1967). In most of these sediments,
the apolar fractions are dominated by the $C_{29}$ *n*-alkane, followed by a high abundance of the $C_{27}$ and then $C_{31}$ *n*-alkanes
(Figs. 2a), such that the ACL ranges from 27.7 to 29.7 with a mean of $29.0 \pm 0.3$ σ (Fig. 5a; Table S1). This ACL (Fig. 5a)
and the dominance of the $C_{29}$ *n*-alkane (Fig. 3a, 4a) across the section suggest that the vegetation at this site was likely
dominated by woody gymnosperms and angiosperms (Bush and McInerney, 2013). Specifically, the ACL of 29.0 is more
likely associated with deciduous rather than evergreen angiosperms, the latter tending to have slightly lower ACLs in
modern vegetation (Bush and McInerney, 2013), although this is not definitive.

The $C_{23}/(C_{23}+C_{31})$ *n*-alkane ratios show some fluctuations throughout the section, suggesting changes in vegetation
type (Fig. 5c; Table S1). For the whole section, the mean average is $0.4 \pm 0.2$ σ and the range is 0.0 to 0.9, spanning almost
the entire theoretical range of 0.0 to 1.0. The $C_{23}/(C_{23}+C_{31})$ values < 0.1 are considered indicative of low inputs from moss
and macrophyte and values > 0.7 are considered indicative of high inputs from moss and macrophyte, particularly *Spaghnum*
mosses due to their particularly high abundance of $C_{23}$ relative to $C_{31}$ *n*-alkanes (Bush and McInerney, 2013). There is
greater fluctuation at the bottom of the section (0-73 m), where values rapidly change across the span of 0.1 to 0.9 values,
indicating dynamic vegetation and environmental change. The upper part of the section (73-340 m) has generally lower
$C_{23}/(C_{23}+C_{31})$ values, averaging $0.2 \pm 0.1$ σ with a range of 0.0 to 0.4, which indicates less frequent wetland-type deposition.
However, these fluctuating values throughout the section that commonly exceed 0.1 indicate that the environment remained
dynamic throughout.

Further providing insights into vegetation type at this site, many sediments also contained diterpenoids and
triterpenoids (Fig. 3a; Table S2; Fig. S3, Fig. S4), compounds indicative of gymnosperms and angiosperms, respectively
(Otto and Wilde, 2001; Diefendorf et al., 2012). Fig. 3A shows an example of a chromatogram with highly abundant
terpenoids (presence, abundances, and their retention times), Figs. S3 and S4 shows the molecular structures of the
terpenoids identified at this site, and Table S2 shows the relative abundance (i.e., absent, trace, present, or abundant) of each
terpenoid at each sample depth. Seven diterpenoid biomarkers associated with gymnosperms were identified throughout the
section (i.e., cadalene, norpimerane, 18-norabietane, 19-norabieta-8,11,13-triene, dehydroabietane, 10,18-bisnorabieta-
5,7,9(10),11,13-pentaene, and simonellite), with notably high abundances from 6.7-8.2 m, 18.1-65 m, and 202.0-301.9 m.
Trace amounts were found throughout most of the section (Table S2). In addition to their association with gymnosperms, the
specific diterpenoid compounds identified at Lühe Basin are common among all major conifer groups and comprise a
particularly high percentage of total diterpenoids in Pinaceae (Diefendorf et al., 2019). The possibility that these diterpenoids
are conifer-derived is especially supported by the presence of simonellite, a compound found in conifer resin (Simoneit,





1986), that was consistently found in the high abundances (often dominant compound) throughout the entire section. Several samples also contained the triterpenoids tetramethyl-octahydrochrysene and Des-A-lupane, compounds synthesized by nearly all angiosperms (Trendel et al., 1989), with notably high abundances from 18.1-46.5 m, 96.0-129.0 m, and 268.0-301.9 m. Trace amounts were found throughout most of the section (Table S2). The more frequent abundance of diterpenoids compared with triterpenoids in these sediments (Table S2) suggest that this environment was likely dominated by gymnosperms with some angiosperms, although it should be noted that taphonomic processes can skew plant preservation and associated biomarker distributions (Tang et al., 2020).

Our biomarker-based vegetation reconstruction is consistent with the plant fossil assemblage recovered from the nearby Lühe town section (which likely corresponds to the 70 to 130 m interval of our coalmine section, see above). Based on high ACL values, the terpenoid interpretation, and variation in $C_{23}/(C_{23}+C_{31})$ ratios, our findings suggest variable vegetation inputs comprising mostly woody gymnosperms (likely conifers), some angiosperms, and very brief periods of *Sphagnum*/macrophyte input (likely associated with localised peatland formation). Correspondingly, the palaeobotany analyses at the Lühe town section assign 38 floral genera to 26 angiosperms, 6 gymnosperms, and 4 ferns (Tang et al., 2020), which were dominated by *Pinus* (i.e., pine) from the family Pinaceae and *Quercus* (i.e., oak) from the family Fagaceae (Xu et al., 2008; Zhang et al., 2020). Palaeobotany analyses also provide evidence of tree stumps, fallen logs, and branches (Yi et al., 2003; Deng et al., 2022). Similar vegetation was described by the palynological assemblages from the Lühe town section, where evergreen oaks (*Quercus*) and alder (*Alnus*) were identified; palynomorphs were dominated by *Quercoidites* (43%), *Titricolpites* (13%), *Pinuspollenites* (7%), and *Piceapollis* (0–20%) (Tang et al., 2020). Palynological findings are not necessarily representative of *in situ* assemblages given that pollen may be blown/washed into the basin from the surrounding (and possibly higher elevation) areas; that said, they are indeed consistent with our biomarker-based assemblages in the Lühe Basin.

Together, the results of the biomarker, palaeobotany, and palynology assemblages indicate a temperate forest with evergreen broadleaved taxa and conifers, and some deciduous broadleaved taxa.

## 3.3 Depositional environment reconstruction

A full and detailed interpretation of the sediment log from the coalmine section can be found in the Supplement (Table S3; Fig. S5); here, we provide only a broad overview, integrating sedimentological and biomarker observations. The measured ca. 340-m thick profile comprises alternations of organic-rich marls, mudstones, sandstones, and lignite deposits, representing various depositional environments typically found in a floodplain setting: active and abandoned channel deposits, proximal to distal overbank crevasse splay deposits, and sub-aerial soils to shallow pond swamps.

The marked variation in sedimentary facies throughout the section suggests it documents a dynamic environment. Similarly, the TOC(%) ranges from 0.1 to 63.9% (Fig. 4a), from organic-lean to essentially maximum (solely organic carbon) values, further indicating a highly variable depositional environment. This interpretation is also consistent with brGDGT-based pH values that vary between 3.0 and 7.7 pH (Fig. 4b; described in more detail below) and with the dramatic





changes in biomarker distributions. For example, $P_{aq}$ values span from 0.0 to 0.9 (representing terrestrial to aquatic-dominated OM, respectively), nearly the entire mathematical range of 0.0 to 1.0, with most values between 0.2 and 0.5 and a mean of 0.4 ± 0.2 σ (Fig. 5b). $P_{aq}$ values that are < 0.23 are considered indicative of terrestrial plant waxes, whereas values that are > 0.48 are common for submerged and floating macrophytes (Ficken et al., 2000). Because $P_{aq}$ sits in the middle of these two key ranges for much of the section, the depositional environment likely had input from both terrestrial and aquatic

sources; this may have been a wet terrestrial environment, like a floodplain, wetland, or shallow lacustrine environment. This wetness is further supported by the sedimentary succession (Fig. 5) and high abundance of *Equisetum* cf. *pratense* in this section (Zhang et al., 2007), a fern species that is indicative of wet terrestrial environments.

  Despite this variability throughout the section, there seems to be a marked difference between the lower (0-73 m) and upper part (73-340 m). The lower part of the section (0-73 m) alternates between swampy conditions and open flood

basin conditions, characterized by fine-grained, typically muddy, sediments, as well as intervals with very high organic matter contents and subaerial exposed and oxidized rooting. This interpretation is supported by the frequent occurrence of high TOC(%) sediments (defined here as > 20%; Fig. 4a), high $P_{aq}$ values, low reconstructed pH (see below), and high $C_{23}/(C_{23}+C_{31})$ *n*-alkane ratios, suggesting that this section was occasionally a peat-forming floodplain environment. The frequent occurrence of high TOC(%) is noted by seven samples in the much shorter 73 m lower section, as compared with

only four in the following 250 m of the upper section. Likewise, $P_{aq}$ values in the lower section range from 0.1 to 0.9 with a mean of 0.4 ± 0.2 σ, whereas the upper section has significantly lower and less variable $P_{aq}$ values ranging from 0.0 to 0.4 with a mean of 0.2 ± 0.1 σ, again indicating a wetter lower section.

  Within this lower section, two sediment intervals (26.7 and 58.5 m) have notably different apolar biomarker distributions from the rest of the Lühe Basin coalmine section more than 2 σ outside the mean distribution (Fig. 5). The

ACLs of 27.7 (26.7 m) and 28.1 (58.5 m) are 2 σ lower than the mean 29.0 ± 0.3 σ (Fig. 5a), $P_{aq}$ of 0.9 (26.7 m) and 0.8 (58.5 m) are 2 σ higher than the mean 0.3 ± 0.2 σ (Fig. 5b), and the $C_{23}/(C_{23}+C_{31})$ of 0.9 (26.7 m) and 1.0 (58.5 m) are 2 σ higher than the mean 0.4 ± 0.2 σ (Fig. 5c). Although these two sediment depths still contain *n*-alkanes with a strong odd-over-even preference and long chain-lengths associated with higher plants (i.e., $C_{27}$, $C_{29}$, and $C_{31}$), they have a clear $C_{23}$ and $C_{25}$ dominance, which is typical for either *Sphagnum* peat mosses (Nott et al., 2000) or aquatic plants (Ficken et al., 2000),

suggesting a very wet environment. Further supporting this interpretation, the low 3.8 pH estimate for the 26.7 m interval is the second lowest pH value for the entire coalmine section, consistent with the low pH commonly found in *Sphagnum* peat bogs (Clymo, 1984).

  Based on all evidence, this lower 73-m interval appears to represent deposition in a relatively low energy fluvial and lacustrine (e.g., meander cut off) system, with fine-grained clay and silt deposition in flood plains, interbedded with lignites

deposited in wet depositional environments, such as a flood basin, wetland/peatland, or shallow lake.

  The upper part of the section (73-340 m) is characterized by coarser-grained deposits representing small influxes of silt and fine sand, occasional rafted branches and leaves, distal crevasse splay features, and river transported wood fragments. At around 69 to 77 m, there is a gradual increase in the energy of the system, with increasingly coarse sediments.





At 74.2 m, a major sand incursion brings in branches/logs, and crossbedding suggests lateral sand migration in a channel, although there is no evidence of basal erosion. At this point, the upper part of the section generally comprises higher energy fluvial environments, represented by e.g., log jams (at 74.2 m, 150 m, 187 m, 224 m, 250-271.5 m, 300.5-316.8 m, 334-336 m), inferred overbank incursion (at 85.8-97.9 m, 271.5-300.5, 315.1 m, 323.4-324.1 m, 331-340 m), and a thick sand body with crossbedding and leaf debris (at 101.3-105.5 m, 233.3-250 m, and 271.5-300.5 m). These are interspersed with some short intervals of swampy conditions, represented by iron horizons, sub-aerially exposed floodplain silts colonised by plants, and four organic-rich lignites. As opposed to the calm flood basin, wetland, or possible shallow lake in the lower part of the section, the upper part of the section suggests a much more dynamic environment, with deposition fluctuating between channel and floodplain. This interpretation supports the (albeit much lower resolution) overview of the Lühe coalmine section described in Wissink et al. (2016).

The biomarker assemblages support this interpretation of the upper section representing a more dynamic environment, with deposition fluctuating between channel and floodplain. The $P_{aq}$ values are still quite variable, ranging from 0.0 to 0.4 with a mean of $0.2 \pm 0.1$ σ, but relatively lower and less variable compared with the lower section. These $P_{aq}$ values suggest a dynamic system dominated by (allochthonous) higher plant inputs over aquatic plants. High variability in pH further supports the interpretation of a dynamic upper section, where the lowest reconstructed pH (3.0 at 303.5 m) and highest reconstructed pH (7.7 at 311.0 m) values in the whole section occur nearly back-to-back, indicating a very rapid shift from a highly acidic (wetland?) environment to a neutral pH setting, likely from the influx of freshwater.

Taken together, our observations indicate a dynamic but evolving fluvial-lacustrine environment throughout the entire section, evolving from a lower energy flood basin, wetland/peatland, and/or shallow lake to a higher energy floodplain and/or channel. We detect abundant terrestrial biomarkers (e.g., leaf waxes, terpenoids indicative of woody gymnosperms and angiosperms, and soil bacterial lipids), consistent with palaeobotanical and palynological evidence in the nearby Lühe town section (Tang et al., 2020). Together, these evidence indicate that the Lühe area was covered in deciduous and evergreen broad-leaved mixed forests. We also see evidence for this being a very wet environment, as indicated by e.g., variable $P_{aq}$ and $C_{23}/(C_{23}+C_{31})$ ratios. However, we do not see strong evidence for this being a deep lacustrine environment which could be indicated by e.g., abundant algal biomarkers and isoprenoidal GDGTs. Instead, we interpret this as a dynamic fluvial system – with the myriad of depositional environments that entails.

**3.4 Climate reconstruction using brGDGT calibrations**

In the fifty-six samples analysed, thirty-eight yielded sufficient brGDGTs for temperature and pH reconstruction (Fig. 6; Table S1). MBT'$_{5me}$ values range from 0.4 to 0.7 with a mean of $0.6 \pm 0.1$ σ; these values remain the same throughout the section. Values in the lower section (0-73 m) and upper section (73-340 m) are nearly identical, respectively ranging from 0.4 to 0.7 with a mean of $0.6 \pm 0.1$ σ and ranging from 0.4 to 0.7 with a mean of $0.6 \pm 0.1$ σ. Over the section, CBT$_{peat}$ values range from -2.1 to -0.2 with a mean of $-1.0 \pm 0.4$ σ. Unlike MBT'$_{5me}$, CBT$_{peat}$ does yield differences between the lower and upper sections, respectively ranging from -1.7 to -0.9 with a mean of $-1.2 \pm 0.3$ σ and ranging from -2.1 to -0.2 with a mean





of -0.9 ± 0.4 σ. This variability is due to the increase in 6-methyl brGDGTs, which corresponds with higher pH. Indeed, pH calculated from $CBT_{peat}$ for the whole section ranges from 3.0 to 7.7 pH with an average of 5.5 ± 0.9 σ; the lower section ranges from 3.8 to 5.9 pH with a mean of 5.1 ± 1.0 σ, and the upper section ranges from 2.97 to 7.7 pH with a mean of 5.7 ±

1.0 σ.

Previous work has proposed a wide variety of brGDGT calibrations for different depositional settings, including lacustrine, soil, and wetland calibrations. Given the variability between these calibrations and the inferred behaviour of brGDGT in different settings, it is perhaps unsurprising that brGDGT indices exhibit large variability throughout the section. This makes it challenging to assign a calibration *a priori*, especially when brGDGTs are not always produced in the

depositional setting in which they are found. Therefore, we initially apply three different brGDGT-temperature calibrations that reflect the depositional environmental variability within the Lühe Basin (Fig. 6).

Most sediments (n=46) are mudstones to sandstones and have a TOC (wt%) ranging between 0.1–23 % with the majority <3 % (Fig. 4). Sandy sediments did not contain sufficient brGDGTs for analysis, but the brGDGTs in the other horizons (n=33) could derive from either allochthonous input from surrounding mineral soils or *in situ* 'lacustrine'

production. Thus, we apply both the soil-calibrated mean annual air temperature ($MAAT_{soil}$; Naafs et al., 2017a) and lake-calibrated MAAT for months above freezing ($MAF_{lake}$; Martínez-Sosa et al., 2021) (Fig. 6). We note that here our MAF values are likely equivalent to MAAT because this site appears to have no months below freezing, as indicated by the model results with cold-month mean temperature (CMMT) of 12.1°C (detailed in Sect. 3.5; Table 2) and palaeobotany-based Climate Leaf Analysis Multivariate Program CMMT of 4.5°C (detailed in Sect. 3.6; Table 3). The remaining sediments

(n=5) have TOC(%) ranging between 40–63 %; these high TOC contents are most likely to have been deposited in a wetland setting and for those, we thus apply the peat-specific MAAT calibration ($MAAT_{peat}$; Naafs et al., 2017b) (Fig. 6).

The overall record (regardless of calibration) exhibits a persistent degree of variability without a long-term warming nor cooling trend. The trends for all three calibrations are virtually the same, but the absolute temperatures differ greatly: the $MAAT_{soil}$ estimates are ca. 9°C cooler than the $MAF_{lake}$ estimates (Fig. 6). $MAAT_{soil}$ values range from 1.9 to 14.4°C, with a

mean of 8.5°C ± 3.5°C σ. $MAF_{lake}$ values range from 11.8 to 22.2°C, with a mean of 17.3°C ± 2.9°C σ. The five $MAAT_{peat}$ values range from 5.4 to 15.4°C, with a mean of 10.9°C ± 4.4°C σ, with values generally closer to those obtained using the soil calibration. The temperature trends throughout the section show variability, possibly due to mixing of *in situ* and allochthonous sources within the rapidly changing and dynamic depositional environment (Sect. 3.3). Given the results and discussion described in Sect. 3.3, the depositional environment is clearly a dynamic fluvial system, and thus the $MAF_{lake}$

calibration appears to be the most appropriate choice for calibration.

**3.5 Climate model results**

To contextualise the climatic and environmental changes occurring at our site, we employed a fully coupled atmosphere-ocean GCM with a range of perturbed Priabonian and Chattian boundary conditions.





First, we tested the impact of $p$CO$_2$, global paleogeography, and site elevation on temperature and precipitation for
the broader Asian region (0°N-60°N, 60°E-120°E). When testing changes from the Priabonian to Chattian (i.e., across the E-O boundary, should our section include this critical boundary), we observe that a decrease from 4x to 2x pre-industrial $p$CO$_2$ results in regional cooling by ca. 6°C, regardless of topographic boundary conditions inferred for the Tibetan Plateau (valley, plateau, or valley-to-plateau; Table 1, simulations 1-3). When assuming no change in $p$CO$_2$, changes from a Priabonian to Chattian paleogeography configuration (which include notable regional changes to gateways such as the retreat of the
Paratethys Sea and the formation of Antarctic ice sheets (e.g., Westerhold et al., 2020)), we observe a slight increase in MAAT of ca. 1.5°C, regardless of topographic changes (Table 1, simulations 4-6). In all model simulations, mean annual precipitation (MAP) increases between 150-200 mm/yr but does not significantly vary amongst different $p$CO$_2$ values nor topographic configurations.

Second, we modelled MAAT and MAP at our site under Chattian boundary conditions, in the Oligocene that our
section certainly spans. Under 2x pre-industrial $p$CO$_2$, the model reproduces a MAAT of ca. 19°C ± 0.4, with seasonal changes varying from a CMMT of 12°C to a warm-month mean temperature (WMMT) of 24°C, regardless of a Tibetan topography with a 2.5-km valley or a 4.5-km plateau (Table 2).

### 3.6 The evolution of the Tibetan region and Eocene/Oligocene climate

Mean annual temperatures across this section are consistent with a temperate climate. Based on the [40]Ar/[39]Ar dating of
feldspars within volcanic ashes exposed at 58 m, which provide ages of 33.32 ± 0.36 Ma (sample lvb11) and 34.38 ± 0.74 Ma (sample lv5 8.0), our Lühe coalmine section may or may not capture the Eocene-Oligocene transition (EOT) that occurred at 33.9 Ma (Fig. 2). We do not find evidence of significant cooling within the first 60 m of the section, the most likely part of our section to have captured the EOT. The lack of cooling could indicate that the Lühe Basin does not span the EOT. Alternatively, our reconstruction may show that Lühe Basin maintained relatively stable temperatures across the EOT
and did not experience the cooling observed at other terrestrial locations (e.g., Zanazzi et al., 2007; Lauretano et al., 2021).

If Lühe Basin does indeed include the EOT, we consider our results in the heterogenous expression of the EOT in terrestrial sections (e.g., Zanazzi et al., 2007; Hren et al., 2013; Sheldon et al., 2016; Lauretano et al., 2021). Terrestrial temperature records across this interval are derived from a variety of qualitative and quantitative proxies (e.g., palaeobotanical, palynological, geochemical). Vegetation records provide the most extensive global dataset of changes
across the EOT and generally show a variety of responses, partly influenced by local/regional factors and changes in precipitation (Pound and Salzmann, 2017). Palaeobotany assemblages from Argentina indicate a 'quasi-static' climate across the EOT (Kohn et al., 2015), whereas assemblages from North America suggest protracted cooling from the early into the middle Oligocene (Retallack et al., 2004). A palynological record from a lignite sequence in SE Australia also suggests a (qualitative) cooling across the EOT; in the same coeval facies, the organic geochemical brGDGT-based record shows 2.4°C
cooling (Lauretano et al., 2021). Terrestrial geochemical records likewise depict a range of responses. Paleosol records from North America, Argentina, and Spain suggest that temperatures remained unvaried during this time (Retallack, 2007;



Sheldon et al., 2012; Kohn et al., 2015), whereas another paleosol record from North America suggests ca. 2-3°C cooling (Retallack, 2007). Geochemical records from the clumped isotopic composition of freshwater gastropod shells from the UK indicate a more intense 4-6°C cooling from the late Eocene to the early Oligocene (Hren et al., 2013). Similarly, the stable

hydrogen isotopic composition from volcanic glass suggests that a 5°C cooling occurred in Argentina (Colwyn and Hren, 2019). Differing still, the oxygen isotopic composition of fossil teeth in North America suggests a dramatic ca. 8°C temperature drop across the transition (Zanazzi et al., 2007). The terrestrial signal during this time certainly shows an unresolved heterogenous response.

If the Lühe Basin coalmine section does indeed include the EOT, then our brGDGT-based records support the

'quasi-static' climate that has gained traction across palaeobotany and paleosol records (Retallack, 2007; Sheldon et al., 2012; Kohn et al., 2015). It should be noted that a relatively muted cooling of < 1-2°C might be difficult to detect in our proxy records, which are better suited for greater temperature oscillations (Naafs et al., 2017b). That said, the lack of significant temperature change over ca. 8 Myr in the Lühe Basin coalmine section is notable (Fig. 6). The results from this section may represent one more important puzzle piece in the terrestrial expression of the EOT (e.g., Pound and Salzmann,

2017) and the possible influence of local factors on this response.

Regardless of the inclusion/exclusion of the EOT, the coalmine section at Lühe basin certainly spans the Oligocene, where our biomarker-, palaeobotany-, and model-based temperature estimates support stable long-term temperatures (Fig. 6). Our brGDGT $MAF_{lake}$ estimates of 17.3°C are not dissimilar to our Chattian model-based MAATs of 19.0°C, CMMT of 12.0°C, and WMMT of 24.0°C (Table 2) nor to the nearby Lühe town section palaeobotanical-based MAATs of 14.5-15.5°C

(bioclimatic analysis; Tang et al., 2020) and MAAT of 16°C, CMMT of 4.5°C, and WMMT of 26.9°C (CLAMP; Table 3). The modelling and palaeobotanical results demonstrate a modest annual range of temperatures, with likely infrequent winter frosting and warm summers. This would suggest a warm temperate climate rather than fully subtropical climate, with taxa that have frost sensitive leaves that are prone to winter deciduousness. CLAMP-based precipitation during the growing season suggests averages of 2250 mm ± 640 σ, whereas precipitation during the three consecutive wettest months (3-WET)

and the three consecutive driest months (3-DRY) range between 1110 mm ± 400 σ and 340 mm ± 98 σ, respectively (Table 3). However, the overall precipitation is likely overestimated in CLAMP, particularly for the dry months in warm climates because water is not a limiting growth factor for plants growing near aquatic depositional sites (Spicer et al., 2011). Notably, the average temperatures for these different methods are higher than our average $MAAT_{soil}$ of 8°C, further supporting that the $MAF_{lake}$ calibration is the most appropriate method for this location and that brGDGTs where likely produced *in situ*

within the water column, consistent with the aquatic plant-dominated depositional environment described in Sect. 3.3.

Importantly, these four different methods for estimating palaeotemperatures match the present-day MAAT for this site. The lack of temperature change from the Oligocene to today suggests that this location has been at its present-day elevation since at least the early Oligocene, supporting the hypothesis that local uplift had already taken place by this time (Spicer et al., 2020; Wei et al., 2022) and supporting the recent work by Wu et al. (2022) who likewise suggest that Lühe

Basin town section had reached its present elevation by the early Oligocene. The modelling results are critical in further



supporting this hypothesis. Topographic features e.g., valley, plateau, or change from a valley to plateau (Table 1) have virtually no impact on the larger climate of this region.

However, we do see dramatic changes in the depositional environment throughout this section, particularly the changes in hydrology and overall energy of the system. Multiple lines of evidence show that there was no significant change

in climate at this site during this time, which points to other reasons for the dynamic depositional environmental changes: topographic changes upstream from the Lühe Basin, precipitation changes, lateral channel migration, or a combination of these. A combination seems likely. The complex tectonic changes that have been explored for sites farther upstream in Tibet likely contribute to the change we see in this section (e.g., Su et al., 2019a; 2019b; 2020; Spicer et al., 2020), which likely impacted precipitation, and the increased fluvial influence then lead to lateral channel migration or an increase in river

size/energy spilling across the basin due to a change in catchment size or the amount of precipitation. Ultimately, untangling topographic changes, precipitation changes, lateral channel migration, or a combination cannot be determined from one site. Determining whether there are synchronous regional upstream changes in catchment characteristics would require a multi-site comparison through the entire depositional basin. Such a multi-site comparison would be a vital contribution to understanding the co-evolving climate and tectonics at this time and recommend this multi-site approach for future research.

**4 Conclusion**

We reconstructed paleoclimatic and paleoenvironmental conditions at the Lühe Basin coalmine section in central Yunnan, China, on the SE margin of Tibet, in the late Paleogene using organic geochemical tools, sedimentology, and climate modelling. Our (primarily) plant- and bacteria-derived biomarkers indicate that this site represented a dynamic environment, likely a floodplain, with occasional submerged peat/swamp deposits and occasional high energy riverine input that increased

in frequency over time. The abundance of terrestrial biomarkers, which indicate woody gymnosperms (likely conifers) and angiosperms, is consistent with previous palaeobotany reconstructions of this area as covered by deciduous and evergreen broad-leaved forests. Temperatures reconstructed from the brGDGT lake calibration indicate a wide range of values (ca. 12-22°C) but the overall average of 17°C across the section is consistent with model-based temperature estimates of 19°C (with CMMT of 12°C and WMMT of 24°C) and palaeobotany-based proxies from the nearby Lühe town section (bioclimatic

analyses estimates of 15°C and CLAMP of 16°C, with CMMT of 5°C and WMMT of 27°C). The temperature obtained from multiple independent lines of evidence, as well as additional evidence from the modelling experiments, shows that past temperatures were similar to present day, indicating that this site has likely been at its current elevation since (at least) the early Oligocene.

**Data availability**

All data is provided with this manuscript.



**Author contributions**

RAS, TS, ZKZ, SFL, PJV, and RDP planned and funded the field campaign. CRW, VL, and JPM conducted the organic geochemistry analyses. AF conducted the model experiments. SHL and HT conducted the palaeobotany interpretations. CRW and VL interpreted the data and wrote the manuscript with contributions from all authors.

**Competing interests**

The authors declare that they have no conflict of interest.

**Acknowledgements and Financial support**

This research was carried out with funding from the joint UK-China Project administered by the Natural Science Foundation of China Project (No. 41661134049, 42072024, 41772026) and the UK Natural Environment Research Council (NERC;
NE/P013805/1). We also thank the NERC for partial funding of the National Environmental Isotope Facility (NEIF; No. NE/V003917/1) that enabled HPLC-MS capabilities. We thank the European Research Council under the European Union's Seventh Framework Programme (FP/2007-2013) and European Research Council Grant Agreement (No. 340923) that enabled GC-MS capabilities. RAS was funded by an XTBG Visiting Scholarship and BDAN by a Royal Society Tata University Research Fellowship. We thank F. Sgouridis at the University of Bristol for technical assistance. We further thank
J. Sepúlveda, D. Fox, and anonymous reviewers for their valuable comments, which greatly improved this manuscript.

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



**Table 1. Climate model simulations (sim.) of the Asian regional impact** with Eocene (Priabonian) and Oligocene (Chattian) boundary conditions are used to test the response of $p$CO$_2$ and Tibetan topography configuration on temperature and precipitation. Grey indicates which parameters are included in the simulation. $p$CO$_2$ is represented as either a change from 4x to 2x pre-industrial $p$CO$_2$ or no change in $p$CO$_2$. Tibetan topography is configured as only a valley, as only a plateau, or as a change from valley to plateau (V-to-Pl). The response is shown on the right as change in mean annual air temperature (ΔMAAT, °C) and change in mean annual precipitation (ΔMAP, mm/yr).

| Sim. | $p$CO$_2$ | Valley | Plateau | V-to-Pl | ΔMAAT | ΔMAP |
|------|-----------|--------|---------|---------|-------|------|
| 1 | 4x to 2x | Yes | No | No | -6.0 | 150 |
| 2 | 4x to 2x | No | Yes | No | -6.0 | 198 |
| 3 | 4x to 2x | No | No | Yes | -6.2 | 182 |
| 4 | No change | Yes | No | No | 1.4 | 167 |
| 5 | No change | No | Yes | No | 1.7 | 173 |
| 6 | No change | No | No | Yes | 1.2 | 199 |





**Table 2. Climate model simulations with different Tibetan topographies (2.5 km valley or 4.5 km plateau) for the Lühe Basin during the Chattian.** Conditions: 2x pre-industrial pCO2 (560 ppm), latitude and longitude (21.1, 101.2), rotated latitude and longitude (29.9, 96.9). Abbreviations: experiment code (expt.), terrestrial lapse rate (Terr-Lapse), mean annual air temperature (MAAT, °C), warm-month mean temperature (WMMT, °C), cold-month mean temperature (CMMT, °C), and mean annual precipitation (MAP, mm/yr).

| Tibetan topography | expt | Terr-Lapse | MAAT | WMMT | CMMT | MAP |
|---|---|---|---|---|---|---|
| 2.5 km valley | tfgkb | 3.67 | 19.0 | 24.8 | 12.12 | 2.63 |
| 4.5 km plateau | tfgkd | 4.83 | 18.9 | 24.5 | 12.15 | 2.73 |





**Table 3. Climate Leaf Analysis Multivariate Program (CLAMP) climate estimates based on the Lühe town section leaf flora and analysed using the PhysgAsia2/Worldclim2 calibration.** For more details on these metrics and how they are obtained see (Spicer et al., 2020b). Row 1: Temperature-related parameters: mean annual air temperature (MAAT, °C); warm month mean air temperature (WMMT, °C); cold month mean air temperature (CMMT, °C); mean minimum temperature of the warmest month (MinT.W, °C); mean maximum temperature of the coldest month (MaxT.C, °C); thermicity i.e., a measure of cumulative heat (Therm). Row 2: Humidity and enthalpy-related parameters: relative humidity (RH, %); specific humidity (SH, g/kg); moist enthalpy (Enth, kJ/kg). Row 3: Vapour pressure deficit parameters: mean annual vapour pressure deficit (VPD.ann, hPa); mean winter vapour pressure deficit (VPD.win, hPa); mean spring vapour pressure deficit (VPD.spr, hPa); summer vapour pressure deficit (VPD.sum, hPa); autumn vapour pressure deficit (VPD.aut, hPa). Row 4: Precipitation and evapotranspiration-related parameters: precipitation during the three consecutive wettest months (3-Wet, cm); precipitation during the three consecutive driest months (3-Dry, cm); mean annual potential evapotranspiration (PET.ann, mm); mean monthly potential evapotranspiration during the warmest quarter (PET.wrm, mm); mean monthly potential evapotranspiration during the coldest quarter (PET.cld, mm). Row 5: Growth-related parameters: length of the growing season i.e., time when the mean temperature is > 10°C (LGS, months), growing degree days > 0°C (GDD0); growing degree days > 5°C (GDD5); growing season precipitation (GSP, cm); mean monthly growing season precipitation (MMGSP, cm).

| Temperature-related parameters | | | | |
|---|---|---|---|---|
| **MAAT (°C)** | **WMMT (°C)** | **CMMT (°C)** | **MinT.W (°C)** | **MaxT.C (°C)** |
| 15.9±2.4 | 26.8±2.9 | 4.6±3.5 | 23±2.9 | 10.4±3.5 |

| Humidity and enthalpy-related parameters | | | | |
|---|---|---|---|---|
| **RH (%)** | **SH (g/kg)** | **Enth (kJ/kg)** | | **Therm (°C)** |
| 65±10 | 8.3±1.8 | 321±0.8 | | 295±75 |

| Vapour pressure deficit parameters | | | | |
|---|---|---|---|---|
| **VPD.ann (hPa)** | **VPD.win (hPa)** | **VPD.spr (hPa)** | **VPD.sum (hPa)** | **VPD.aut (hPa)** |
| 6±2.4 | 3.2±1.5 | 4.7±4 | 8.7±3.5 | 7.4±2 |

| Precipitation and evapotranspiration-related parameters | | | | |
|---|---|---|---|---|
| **3-Wet (cm)** | **3-Dry (cm)** | **PET.ann (mm)** | **PET.cld (mm)** | **PET.wrm (mm)** |
| 111±40 | 35±10 | 1002±166 | 27.5±14 | 125±24.5 |

| Growth-related parameters | | | | |
|---|---|---|---|---|
| **LGS (month)** | **GSP (cm)** | **MMGSP (cm)** | **GDD0** | **GDD5** |
| 9.8±1.1 | 225±64 | 24±7 | 677±118 | 735±106 |



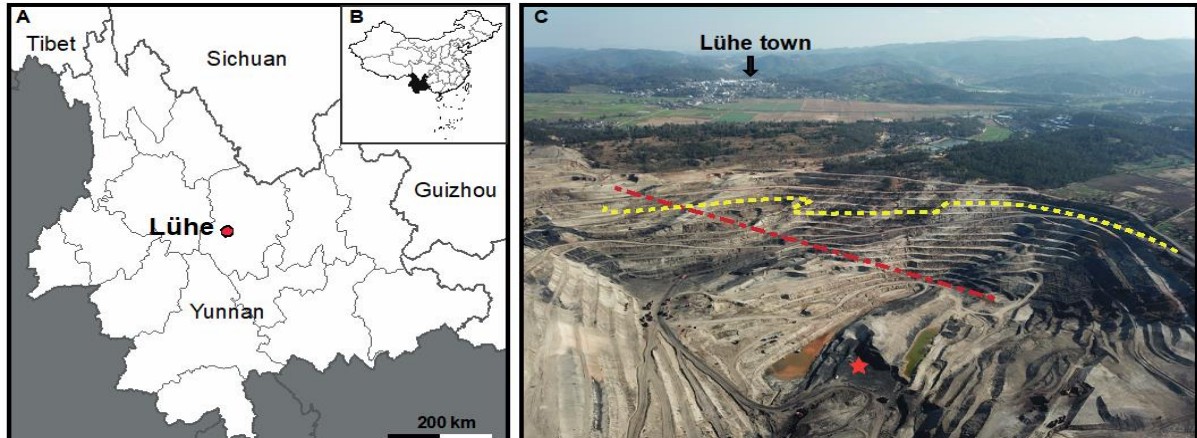

715 **Figure 1: Location and overview of the Lühe coal mine section.** A-B: Location map (25°10′N, 101°22′E). C. Photograph of Lühe Basin coalmine section (yellow line indicated the sampling log of this study, red indicates the section logged by Li et al., 2020). Lühe Basin town section, noted in C, is located 2.6 km away.





**Figure 2: Geologic age and simplified section log for Lühe Basin.** (a) Age in millions of years (Ma) alongside the geological timescale by series and stage, and updated geomagnetic polarity timescales 2020 (Speijer et al., 2020). (b) Geomagnetic polarity for Lühe Basin reported in Li et al. (2020) and associated Chrons. The grey lines connect the global magnetostratigraphy with this basin. (c) Simplified section log for Lühe coalmine section. Star marks $^{40}Ar/^{39}Ar$ dating. Detailed sediment log can be found in Fig. S5 and Table S3.



**Figure 3: Total ion chromatograms of the apolar fraction.** (a) Depth from base 268.0 m with high content of terpenoids and *n*-alkanes exemplary of the section, especially the $C_{29}$ *n*-alkane dominance. (b) Depth from base 58.5 m exemplary of the two outliers with $C_{23}$ and $C_{25}$ *n*-alkane dominance. Numbers represent: 1. Cadalene, 2. Norpimerane, 3. 18-norbietane, 4. 18-norabieta-8,11,13-triene, 5. Dehydroabietane, 6. 10,18-Bisnorabieta-5,7,9(10),11,13-pentaene, 7. Naphtalene, 8. Simonellite, 9. TetramethyI-octahydrochrysene. Gold boxes zoom in on m/z 191 i.e., hopanes used for the thermal maturity index.





**Figure 4: Indicators for thermal maturity.** (a) Percentage of total organic carbon (TOC) shown with green triangles. Note: scale is condensed above 10%. Gray dotted line shows high TOC samples. (b) pH show with blue squares. (c) Hopane isomerisation index of $\beta\beta/(\beta\beta+\alpha\beta+\beta\alpha)$ for $C_{29}$ (lavender circles) and $C_{31}$ (red diamonds) hopane. Red dotted line showing 2-pt moving average. Gray dotted line shows the threshold for (general) mature versus immature values. (d) Carbon preference index shown in yellow squares. Yellow dotted line showing 2-pt moving average. Gray dotted line shows the threshold for (general) mature versus immature values.





**Figure 5: Indicators for depositional environment.** (a) Average chain length (ACL), an indication of organic matter sources, shown in lime squares. (b) $P$-aqueous ratio ($P_{aq}$), an indication of wetness, shown in blue circles. Gray dotted lines indicate interpretation for terrestrial plants (below 0.23) and submerged and floating macrophytes (above 0.48). (c) $C_{23}/(C_{23}+C_{31})$ $n$-alkane ratio, an indication of ecology, shown in green triangles. Gray dotted lines indicate woody angiosperms (below 0.1) and *Sphagnum* mosses (above 0.7).





**Figure 6: brGDGT reconstruction of pH and temperature.** Temperature from calibrations based on possible depositional environments: mean temperature for month above freezing using a lacustrine calibration (MAF$_{lake}$ in lavender circles; Martínez-Sosa et al., 2021) and MAAT using a peat calibration for samples with total organic matter >30 % (MAAT$_{peat}$ in black squares; Naafs et al., 2017). Our model-based temperature for the Chattian is shown in red. For comparison, temperature estimates from bioclimatic analyses (BA) light green and Climate Leaf Analysis Multivariate Program (CLAMP) in dark green, including cold month mean temperature (CMMT), mean annual air temperature (MAAT), and warm month mean temperature (WMMT) from the Lühe town section, which represent a single temperature averaged that (should) correspond with ca. 70-130 m in our coalmine section. Light blue shading shows the range for modern day temperatures at this site.