# Peer review of "Dynamic environment but no temperature change since the late Paleogene at Lühe Basin (Yunnan, China)"

_EGUsphere, 2023_

## Author Comment (AC1)

***Reply to reviewer comments 1 (RC1):***

*Witkowski et al. reported multiple biomarker records from a late Paleogene section from the Luhe Basin on the southeastern Tibetan Plateau. The biomarker indicators include TOC, n-alkanes, hopanes, terpenoids, and brGDGTs. The section may or may not include the Eocene-Oligocene Transition (EOT) around 33.9 Ma. Based on these biomarker records, authors infer that regional vegetation did not change significantly over the study period (likely the Rupelian), while the depositional environment changed dramatically from a lower energy flood basin, wetland/peatland and/or shallow lake to a higher energy flood plain and/or channel. The brGDGT-based MBT'5me temperature record shows high variability but does not show particular long-term trending. The authors conclude that the regional temperature has not changed much since the late Paleogene (with the support of model simulations), despite the dynamic changes in depositional environment.*

*The tectonic evolution in the Tibetan region has affected the Asian monsoon system and global climate, as well as biodiversity. However, the links among them is far from being understood. This study, reporting climatic and environmental changes during the Rupelian period from the southeastern Tibetan Plateau, would definitely contribute to our knowledge on this topic, and should be of broad interest to scientific community. This manuscript is worth to be considered by the journal. I provide a few comments below for authors' consideration.*

**We thank the reviewer for their positive comments and their constructive feedback. We respond to each comment below in bold text. Line numbers refer to the "track changes" version of the manuscript.**

*Authors stated that the regional temperature during the Rupelian period was close to present-day value, which is also reflected in the title. Although authors cited paleobotany- and model-based results to support their claim, it is quite difficult to accept this claim As in authors' simulations, regional temperature cooled by ~6C from 4x to 2x preindustrial $pCO_2$ (with topographic effect contributing little), it is hard to imagine the regional temperature would have remained the same when $pCO_2$ dropped from 2X to the preindustrial value. I would suggest that authors might check their 2x simulation to find out why temperature did not change significantly from the preindustrial value and provide such an explanation in their text. If such an explanation is not available, then authors might consider weakening this statement a bit.*

**This is an excellent point. In our description, we were referring to the Rupelian and Chattian, which this section definitely contains; in the Rupelian and Chattian, the palaeobotany, models, and biomarkers all closely agree (e.g., Fig. 6). The 6°C cooling in the models is across the Eocene-Oligocene transition, which we were more tentative in claiming in this section, given the uncertainties on the Ar/Ar dating. We have added a paragraph describing why the model and biomarker data disagree across this boundary (if this section contains the EOT), and that "this difference is likely due to spatial resolution of the model (large homogenous grids) versus the biomarker data (specific details on a basin scale); the model cannot capture the complex topography that may influence the climate at this site." Developing higher spatial resolution models are currently being pursued (but are not available at this time).**

*Authors provided two possibilities that the EOT may or may not be present in their studied section. However, I note that at the bottom of their stratigraphic column, Chron C15n, which is around 35 Ma, appears to be well defined. Authors do not have much confidence in this chron?*

**Based on the reviewer's comments, we now include the ages in all relevant figures and have added ~Line 98, "The uncertainties in the age are important to note, as it means that the Lühe Basin may or may not cover the Eocene-Oligocene transition (EOT) that occurred at 33.9 Ma, though the palaeomagnetism suggests that this section likely includes the EOT."**

*Also, the study period covers the most of Rupelian period, not Chattian period authors claimed (although simulation results for the Chattian period should be OK for comparison with authors' results).*

**We agree. We note that there are minimal differences between the Rupelian instead of Chattian, especially in terms of the relative uncertainties between intra-stage reconstructions. The main point here was to differentiate between the non-ice period hothouse of the end Eocene vs. the icehouse conditions of the Oligocene. We more explicitly refer to the Rupelian results throughout the manuscript.**

*Authors attributed the large variability in MBT'5me, ~0.3 unit which is equivalent to 10-15 C temperature changes to the mixture of in situ vs. allochthonous brGDGTs. Could authors compare the MBT'5me with one of the environmental indicators (for instance, Paq?) to see if a possible relationship could be found? This is quite important as authors stated that depositional environmental changed from earlier to later time. That is, such depositional environment changes could have contributed to/resulted in the no overall trending.*

**This is an excellent suggestion to further test whether the temperature is impacted by changes to the mixture of in situ versus allochthonous brGDGTs. We plotted MBT'5me against Paq (as well as MBT'5me against lithology), but there are no apparent trends.**

*Similarly, authors in one place (around Line 360) mentioned the increase in 6-methyl brGDGTs, which could also be verified by authors' CBT/pH record. Authors might be aware that the increase in 6-methyl brGDGTs, i.e., the change of IR6me value, could significantly affect MBT'5me value, as reported in recent studies, for instance, Wu et al., 2021, CG, https://doi.org/10.1016/j.chemgeo.2021.120348 and Wang et al., 2021, GCA, https://doi.org/10.1016/j.gca.2021.05.004. Authors are strongly suggested to evaluate this effect. My sense is that MBT'5me value could be lowered a bit if this effect is corrected, based on the information authors provided. If correct, authors may not need to present two possibilities whether EOT is present or not (my Point #2), and authors might see relatively high temperature at the lower part, whihc could represent the late Eocene interval. Authors might also want to look at the MBT' index.*

**We agree. We have reworded this section to more accurately reflect the record ~Line 359-363, "This variability in CBTpeat (but not in MBT'5me) is due to four samples in the upper section (depths 273.5, 301.9, 311, and 331.5 m) that contain 6-methyl brGDGTs; these are the only samples that contain 6-methyl brGDGTs in this whole section. Our tests show no correlation of the 6-methyl brGDGTs with overall changes in depositional environment (e.g., against Paq values) but do show expected changes with pH (Wu et al., 2021; Wang et al., 2021). Indeed, the four 6-methyl brGDGT-containing samples have high pH values (respectively, 7.6, 6.3, 7.7, and 6.6 pH), as compared with …"**

**The MBT'5me temperatures with/without 6-methyl samples are similar throughout (the four samples with 6-methyl brGDGTs are respectively 20.4, 14.1, 15.5, and 16.5C). These four samples are the youngest/upper part of the section (latest Rupelian, earliest Chattian) and do not impact the EOT at the bottom/lower part of the section.**

*A recent study He et al., 2022, Science Bulletin https://doi.org/10.1016/j.scib.2022.10.006 might be added to support the view that this region might have reached present elevation since the late Eocene.*

**Added (Line 460).**

*Some explanation should be provided for the lithologic column in Fig. 2.*

**Added.**

*pH reconstruction is not plotted in Fig 6 but stated so in figure caption.*

**Removed from figure caption.**

---

## Author Comment (AC2)

***Reply to reviewer comments 2 (RC2):***

Witkowski et al. reconstructed the temperature and deposition environment at a site in Yunnan, China, using multiple biomarkers. Their reconstruction indicates that the temperature at this location was similar to today at ~33 Ma, which may imply that its altitude might have reached similar height to today. Because this site is in the southeastern margin of the Tibetan Plateau, the results may be of value to people who study the evolution of the Tibetan Plateau.

The part of manuscript that is related to biomarkers is well written but the part related to climate modeling is quite poor. Lots of information are missing. I could not find even a single relevant figure which made it difficult to understand how the experiments were setup. The results of the climate modeling were not used in a meaningful way either, making me wonder why bother carrying out that many model simulations. Detailed questions and comments are listed below.

**We thank the reviewer for their positive comments and their constructive feedback. We respond to each comment below in bold text. Line numbers refer to the "track changes" version of the manuscript.**

1. A figure is certainly needed in order for the readers to understand how the model experiments were set up. For example, what do they mean by a constant valley or plateau (near line 195)? What do the corresponding results tell us?

**We have added a figure (Fig. 5) to demonstrate the different between the valley (2.5km) and plateau (4.5km). We have also specified this more clearly throughout the section. See section 2.2.5, "Topographic changes were also considered across the Priabonian to Rupelian based on hypotheses posed by Spicer et al., 2020 (and references therein), either as a) a constant valley at 2.5 km elevation, b) a constant plateau at 4.5 km elevation, or c) a change from a valley at 2.5 km to a plateau at 4.5 km." We have also added some guiding text in the caption in Table 1, "Tibetan topography is configured at different elevations to determine the impacts this may have one the broader climate system, here showing as only a valley at 2.5 km, as only a plateau at 4.5 km, or as a change from valley to plateau from 2.5 to 4.5 km."**

2. Since the model results show that there should be a large change of annual mean temperature (~6 °C) when CO2 is changed from 4x to 2x, shouldn't they indicate that the section does not include the Eocene-Oligocene Transition (EOT)?
**Based on our age model, the section should include the EOT. Although the model indicates that a CO2 change from 4x to 2x should result in 6C cooling, we do not see that in the data recorded at this site. We have added an explanation of why there might be a difference in the model versus biomarker data on Line 439-444, namely that, "This difference is likely due to spatial resolution of the model (large homogenous grids) versus the biomarker data (specific details on a basin scale); the model cannot capture the complex topography that may influence the climate at this site."**

3. In my opinion, the results are insufficient to claim that the altitude of the site had reached present-day value based on only the surface temperature. Many other factors could impact on the temperature of a local region. For example, did the latitude of the site change much between the Early Oligocene and PI? Was there a large change in the temperature of South China Sea? Was the region more cloudy during the Early Oligocene than in PI?

**The biomarkers and palaeobotany proxies record the temperature that the organisms/assemblages experience in their lifetime, so the reviewer is correct that these methods leave room for alternative factors that could impact our understanding of the temperature at the section (i.e., the altitude was lower, but it was cloudier). However, this explanation seems unlikely given the agreement of the models and supporting proxy data which we have now included more explicitly in this section, ~Line 494, "An alternative explanation may be that there are complicating factors that impact temperature for the biomarker and palaeobotany proxies e.g., cloud coverage. However, this seems less likely given the proxy agreement with the model results. This coupled atmosphere-ocean general**

**circulation paleoclimate model includes paleo-rotations (latitude/longitude), changes in globally dynamic temperatures (e.g., SST changes from the South China Sea), and changes in cloud coverage and precipitation. In addition, there is supporting evidence from moist enthalpy from CLAMP and oxygen isotopes from carbonate nodules that topography of the eastern margin of Tibet was established immediately prior to and during the basin development (e.g., He et al., 2022)."**

4. Some of the co-authors had published climate evolution for the past 100 million years, it should be convenient to look at how the temperature at this site had changed during the past 30 million years. Similar data were also published by Li et al. (2022, Scientific Data, https://doi.org/10.1038/s41597-022-01490-4).

**In this manuscript, we are focused on how this specific site changed during this tectonically and climatically dynamic time in Earth history, and how this specific site fits into our understanding on local, regional, and global scales. By looking at changes in temperature at a specific location, we can gain both spatial and temporal resolution that models are not capable of reaching, given they run at geologic stage level and at 300 km spatial grids. In return, the model offers context that a single site cannot. That is why it is so valuable to combine modelling and data, as we have done here.**

5. The authors may also look at how the precipitation changed at the site from the early to middle Oligocene from the two datasets mentioned above since it is directly related to their reconstruction regarding the deposition environment.

**We have added a paragraph to Sec. 3.6 on precipitation to discuss the results from the model and CLAMP with our interpretation of the sedimentation and organic geochemistry, all of which support a significant hydrological shift from the Priabonian into the Chattian.**

Minor comments

WMMT and CMMT are not defined, are they the temperatures for the warmest and coldest month or the average of a few months. While at lines 445-446, the dry or wet months are defined.

**We have defined CMMT and WMMT for the coldest/warmest month mean temperature for CLAMP, as well as adding 3CMMT and 3WMMT for the three consecutive coldest/warmest month mean temperature for the model.**

L190: downscaled -> upscaled?

**Different scientific communities use this term differently; the modelling community generally use "downscaled" to describe going from a low-resolution (more course representation) to higher resolution (more detail). To avoid confusion, we've removed the phrasing altogether and replaced with "…at low (3.75° x 2.5°) and scaled to the high model resolution (0.5° x 0.5°)…"**

L397: increases -> varies? Otherwise, I do not understand why the precipitation always increases.

**We have added ~Line 408, "…mean annual precipitation (MAP) increases between 150-200 mm/yr from the Priabonian into the Chattian but is not significantly impacted by different pCO2 values (e.g., from 4x to 2x pre-industrial pCO2 or no change in pCO2) nor topographic configurations in the model conditions. This suggests MAP was impacted by global changes across this boundary e.g., the opening of ocean gateways (e.g., Drake Passage; creation of the Antarctic Circumpolar Current), the massive expansion of the Antarctic icesheet, and broad reorganisation of the global climate system (Westerhold et al., 2020)."**

L445: what are the modelled precipitation?

**The resulting modelled precipitation can be seen in both Table 1 under change in mean annual precipitation (delta MAP), and in Table 2 under MAP (mm/yr).**

L449: where -> were

**Changed.**

Figs. 4-6, in all three figures, the Chrons C12n are located above C11n, inconsistent with Fig. 2.

**We have revised the figures to add the age model, based on the comments of Reviewer 1.**

---

## Author Comment (AC3)

**Reply to reviewer comments 3 (RC3):**

In this manuscript, the authors investigated a 340-m long section in the Lühe Basin in Yunnan, China, located along the SE margin of the Tibetan Plateau. Based on analyses of organic geochemical proxies and model simulations, the authors reached several conclusions: (1) the sediments in the Lühe section were deposited in a highly variable and dynamic environment; (2) there was no significant temperature change (no cooling trend) at the study site between 35 Myr and 27 Myr; and (3) their reconstructed temperatures were similar to the present-day, suggesting that the study area has reached its present elevation since at least the early Oligocene. The topic of this study is interesting and important. The investigation of the sediments from the SE margin of the Tibetan Plateau could contribute to a better understanding of the complex tectonic and environmental history of the Plateau. However, there seem some major problems in the manuscript, which makes the conclusions (2) and (3) unconvincing.

**We thank the reviewer for their positive comments and their constructive feedback. We respond to each individual comment below in blue/bold text. Line numbers refer to the "track changes" version of the manuscript.**

In general, there is lack of discussion and information on the limitation and uncertainty of various temperature reconstructions used in this study. It is difficult to judge whether different reconstructions are comparable, whether the past and the present are comparable, and whether the conclusions are reliable. From some information given in the main text and Fig.6, it is difficult to conclude that there is no trend in temperature change over the study period.

**We added more descriptions of the uncertainties associated with each method (Line 465-470). We include the number of samples, $r^2$ value, and associated uncertainty of each brGDGT calibration used throughout the methods (Lines 164, 169, 177, 182). When referring to mean averages within our descriptions of the dataset, we always include the standard deviation.**

**Throughout these comments, the reviewer asks whether the reconstructions are comparable. Indeed, these are all well-established methods that have been tested for decades, both separately and together, both against the modern and against past periods of established climate regimes; again and again, these methods (nearly always) converge on the same exact values, which confirms they are tracking the same climate signal. In our section, our four different methods all converge on similar values, adding to the extensive literature.**

**The different methods work together complementarily, each with its own strengths and limitations, which is why it is so important to use multiple approaches in paleo records. For example, brGDGTs and CLAMP have similar uncertainties (2.4°C and 2.3°C, respectively); however, brGDGTs provide much higher temporal resolution while CLAMP provides contextual information like precipitation and vapour pressure deficit. The data-based methods (e.g., brGDGTs and CLAMP) represent reality, recording actual temperatures from the site during that time period, as opposed to the models which are physics-based expectations; however, the models provide larger scale and contextual views of the region that are impossible to achieve with the proxies alone.**

Some model results are hard to understand (see specific comments below), so using the model results to support the GDGTs-estimated temperature is unconvincing.

**To improve the clarity of the climate model results, we have rewritten the entire climate model results (Section 3.5), added a table to show changes throughout the section (Table 2), and added a new figure to visualise the differences in orographic height in our modelling experiments (Fig. 5). We address each specific comment below.**

Given the many uncertainties and doubts detailed in my comments below, conclusions (2) and (3) are too strong. Moreover, regarding the elevation of the Tibetan Plateau, many other studies suggest that rapid elevation increase in SE Tibet had happened during the Miocene (e.g. Clark et al 2005, doi: 10.1130/G21265.1; Royden et al 2008, 10.1126/science.1155371), which contradicts the conclusion (3).

**Although some previous studies have suggested rapid elevation during the Miocene for SE Tibet, they have since been superseded; many of these original basins that were previously reported as Miocene (based on floral assemblages) have now been radiometrically dated and confirmed as Eocene (e.g., Tang et al., 2020; Xu et al., 2023). This new and very large body of work demonstrates the rise of SE Tibet in the Eocene.**

**That said, this region is indeed complex, which is why the current study provides an important puzzle piece in characterising the heterogeneous rise of SE Tibet.**

My specific comments and questions are listed below:

1. About proxy problems:

- Regarding temperature estimation from brGDGTs, to which extent could the global calibration be applied at the study site and for such an old time period, and what could be the uncertainty and limitation?

**The proxy is based on a biological mechanism / function and global calibration datasets (outlined in Section 2.2.4 e.g., Weijers et al., 2007; De Jonge et al., 2014; Martinez-Sosa et al., 2021, and including members of our team e.g., Naafs et al., 2017) and is thus not dependent on location. This is a widely used proxy for temperature reconstructions across the Cenozoic (e.g., Weijers et al., 2007) that has been supported by its close replication of other deep-time temperature proxies throughout the literature (e.g., Hollis et al., 2019).**

**Within this current study, we see that the brGDGT temperatures are consistent with the temperatures reconstructed from two independent palaeobotany-based proxies (BA and CLAMP) as well as consistent with the climate model simulations. Together, these independent methods provide supporting evidence that these methods do indeed work and this study adds to the extensive literature on cross-proxy comparisons with brGDGTs.**

- It is written that "the GDGTs in our section could have been produced in mineral soils, peats, or a shallow lake environment based on our environmental reconstructions". Then what is the exact meaning of MAAT_soil, MAAT_peat and MAAT_lake? They are assumed to reflect the temperature of the soil, peat and lake water where the GDGTs were produced, instead of air temperature. If the present-day MAAT and the simulated MAAT are air temperature, they would not be comparable with the GDGTs-based temperatures.

**brGDGTs are calibrated against air temperature, which are closely related to mean annual temperatures from soil (De Jonge et al., 2014), mean annual temperatures from peat (Naafs et al., 2017), and months above freezing temperatures from lakes (Martinez-Sosa et al., 2021). These three calibrations are based on extensive studies in the literature, with detailed descriptions in Section 2.2.4.**

- The authors conclude that "our observations indicate a dynamic but evolving fluvial-lacustrine environment throughout the entire section". So to which extent the reconstructed temperature reflects in-situ temperature or temperature at remote source region is uncertain. The comparison with the present-day temperature at the study site is therefore questionable.

**We agree with the reviewer that a dynamic environment could have provided allochthonous input which could bias (but unlikely to overwhelm) the in situ signal. We tested whether the temperature is impacted by changes to the mixture of in situ versus allochthonous brGDGTs by plotting MBT'5me against Paq and plotting MBT'5me against lithology (Line 360-365). There were no apparent trends. This suggests primarily in situ signals. Furthermore, our brGDGTs were only found in the low-energy clay and silt sediments (more likely to contain in situ signals); the high-energy sand sediments (more likely to contain allochtonous signals) were not conducive to brGDGT preservation (Line 390-400). In other words, our brGDGT temperature reconstructions mostly represent in situ conditions.**

- Given the highly variable depositional environment at the study site, it is difficult to understand why only the lake-calibrated temperature is considered. Given the marked difference between the lower

(0-73 m) and upper part (73-340 m), should different calibration methods be used at least for the upper and lower parts? Could this affect the conclusion?

**We have used the lake calibration because the depositional environments are characterised as fluvial-lacustrine (a floodplain, with occasional submerged peat/swamp deposits and occasional high energy riverine input). All of these possible environments throughout the section (both upper and lower sections) are very wet and thus, the lake calibration is the most reasonable calibration to use (See detailed explanation in Lines 390-400).**

- I read from Fig.6 that the uncertainty of the lake MAF is ~8 degree C (shown by blue horizontal bars). Given such a large uncertainty, it is impossible to conclude whether there is a trend or not over the study period. Many studies cited in the paragraph of line 411 suggest a cooling across EOT of only a few degree C. If there was also a cooling trend of a few degree C at the study site, it could not be reflected in the GDGTs temperature reconstruction because of the large uncertainty and large variability.

**The uncertainty associated with the brGDGT lake calibration is actually ± 2.4°C; we made a mistake by using a different calibration's ± 4.0°C which has now been corrected. We thank the reviewer for highlighting this issue.**

**But it is also important to note that this is the standard deviation of the calibration, meaning that the absolute estimated values are the most likely values in a normal distribution and that the likelihood of the temperature being e.g., +2C or -2C is much lower. Although this is a large range, this is considered well-constrained uncertainty for a paleo-proxy (e.g., Hollis et al., 2019). We also want to highlight recent work by members of our team on reconstructing temperature change across the EOT using brGDGTs (Lauretano et al., 2021).**

- The authors state that "The temperature trends throughout the section show variability, possibly due to mixing of in situ and allochthonous sources within the rapidly changing and dynamic depositional environment (line 382)". So the lack of temperature trend over the 8 Myr at the study site could be due to mixing and large variability but not really related to climate change.

**This is not likely, see our previous comment on in situ vs allochthonous sources.**

- As the brGDGT indices exhibit large variability throughout the section and the MAF_lake has a range from 11.8 to 22.2°C, which covers a wide range of possible temperatures, considering only the mean temperature of 17.3°C in the comparison with the present-day temperature and the simulated one does not make sense.

**We include the full range, not just the average, to show readers that there is indeed variability over time. However, it is notable that there is no long-term cooling over the course of the section nor is there a stepwise cooling from the Eocene into the Oligocene, as seen across the marine-based records (and some terrestrial brGDGT based records). This lack of change from the globally warmhouse temperatures of the Eocene into the globally coolhouse temperatures of the Oligocene is certainly interesting and adds to the increasing literature that suggests a heterogeneous signal in the terrestrial realm across the EOT.**

- In Fig.6, there are only five points for peat brGDGT MAAT and no information on MAAT_soil, so it is impossible to get trend information from these two reconstructions. The BA and CLAMP MAAT are only an averaged estimate for a thick portion (70-130m), so they can not indicate trend.

**All raw data is included in the Supplement. The five points for peat brGDGT were included because these five samples that have extremely high organic content and thus may have come from a peat-like depositional environment. The MAAT soil calibration was not used because there was no strong evidence for a soil-dominated depositional environment (see Section 3.3 on depositional environment reconstruction).**

**The reviewer is correct that BA and CLAMP MAAT represent a single point (integrating 18m in the section) and do not indicate a trend. This highlights the benefit of using brGDGTs as these can highlight changes across a section.**

2.   About model results

-   Line 400-402: It is difficult to understand that the simulated temperature at the study site is not sensitive to a large change of the Tibetan topography from 2.5-km valley to 4.5-km plateau. This shows that the temperature at the study site is not sensitive to the change of the Tibetan topography. Then it would not make sense to use the reconstructed or simulated temperature at the study site to indicate the Tibetan elevation.

**We would not necessarily expect a large change in the climate signal simply because of the change in the configuration of Tibet from the proposed topographic sensitivity studies modelled here. This is because the dynamics and thermodynamics are not sensitive to large scale changes in Tibet for regions of the Hengduan, as long as there is high topography in the region already, in this case the Gangdese vs. a Plateau (both at 4.5km). The seminal work of Boos and Kuang (2010) show in the modern world, as long as you have a large topographic front (in this case the Himalaya, analogous to the Gangdese here) and not a Plateau, the monsoon in Asia does not significantly affect the advected South/South-west airmass flow and related temperature and precipitation patterns. Local changes in topography are more important to reconstruct elevation.**

-   Line 397: "In all model simulations, mean annual precipitation (MAP) increases between 150-200 mm/yr but does not significantly vary amongst different pCO2 values nor topographic configurations.": It is difficult to understand that the large boundary condition changes have no significant influence on MAP. It means that in the model, the MAP in the Asian region is not sensitive to these condition changes. Then what could have caused the changes in hydrology and overall energy of the system in the study region (line 458-459).

**We suggest that upstream changes are occurring (Line 500-505): "The complex tectonic changes that have been explored for sites farther upstream in Tibet likely contribute to the change we see in this section (e.g., Su et al., 2019a; 2019b; 2020; Spicer et al., 2020), which likely impacted precipitation. The increased fluvial influence then led to lateral channel migration and/or an increase in river size/energy spilling across the basin due to a change in catchment size or the amount of precipitation."**

-   The simulated CMMT is 12.1°C, and the CLAMP reconstructed CMMT is 4.5°C. Given such a big difference, which result is more reliable, the model or the CLAMP?

**Each method has different strengths and limitations, which is why it is so valuable to use multiple approaches in paleo research. CLAMP is based on fossil assemblages that integrates traits from dozens of species; as a data approach, it is generally more reliable but only for the limited location and time where the data was collected. The model provides powerful largescale spatial and temporal context but lacks the resolution on the basin scale. In Su et al. (2019), we look at the model latitudinal results versus CLAMP, and find a strong regression line between the two.**

-   Line 390: What is the purpose to analyze the results for such a broad Asian region? Its relevance for the current study is unclear. In Table 1, it would be more relevant to show the model results for the Lühe basin instead of the very broad Asian region.

**We have revised Table 2 to focus exclusively on the Lühe Basin site. We have left Table 1 on the regional results, as these are critical for contextualising the site.**

-   Line 186-189: It would be helpful to show a figure with the present-day climate simulation for the Lühe basin and the comparison with observation.

**A present-day climate simulation for the Lühe Basin would simply look like the real present-day Lühe Basin. Valdes et al. (2017) shows a comparison of the HadCM3 model to other models, including the models that only focus on the modern. HadCM3 consistently does among the best. Furthermore, Sperber, et al. (2013) shows the CMIP5 monsoon statistics for Asia between all the models; again, HadCM3 is one of the top models.**

- Line 190: It would be helpful to show the boundary conditions on the model resolution to show to which extent the model could resolve the regional orography condition at the Lühe basin and the SE Tibetan Plateau region.

**All model boundary conditions are included in Supplement Table S4 and S5.**

- It would be helpful to give more explanation on experiment setup and provide significance test for the simulated changes of temperature and precipitation. What does it mean "valley, plateau, v-to-PI", please give more information and illustrate with figures.

**We added a figure (Fig. 5) to demonstrate the different between the valley (2.5km) and plateau (4.5km).**

**We specified this more clearly throughout the section. See section 2.2.5, "Topographic changes were also considered across the Priabonian to Rupelian based on hypotheses posed by Spicer et al., 2020 (and references therein), either as a) a constant valley at 2.5 km elevation, b) a constant plateau at 4.5 km elevation, or c) a change from a valley at 2.5 km to a plateau at 4.5 km."**

**We added some guiding text in the caption in Table 1, "Tibetan topography is configured at different elevations to determine the impacts this may have one the broader climate system, here showing as only a valley at 2.5 km, as only a plateau at 4.5 km, or as a change from valley to plateau from 2.5 to 4.5 km."**

3. About temperature response to CO2 decrease:

The CO2 concentration decreased from 1120 ppm for the Priabonian to 560 ppm for the Rupelian, and the model simulates a cooling of ~6C for the broad Asian region (Table 1) in response to the CO2 decrease. However, based on the GDGTs reconstructed temperature, the authors conclude that there was no significant temperature change (no cooling trend) over the 8Myr period at the study site. It is difficult to understand that such a large CO2 decrease has no significant impact on the temperature at the Lühe basin. With a high elevation, the temperature of this region would be more sensitive to the large CO2 decrease than low-elevated region. Does the model simulate a cooling at the Lühe basin in response to the large CO2 decrease?

**Although there was a CO2 change worldwide and thus can have an impact on temperature worldwide (and on average for the broad Asian region), CO2 is not the only factor that impacts local temperature. Indeed, there appears to be global heterogeneity of terrestrial temperature records during the Eocene-Oligocene transition, as we outline in Section 3.6. For example, impacts may include localised albedo (e.g., based on soil type), vegetation type and associated impacts (e.g., transpiration and canopy cover (Fritz et al., 1961)), localised nutrient and carbon cycling, and detailed differences in topography (e.g., even as detailed as whether the site represents the north or south slope of a valley). This detailed local resolution is not possible with the resolution of the models.**

Other comments:

- As the sediments at the Lühe coalmine section were deposited in a highly dynamic and variable environment that fluctuated between low energy floodplains and high energy fluvial systems, it would be important to prove the continuality of the deposition in the section.

**We have the age framework in our figures, which considers sedimentation, time series analysis, astronomical tuning, and the Ar/Ar dating (Li et al., 2020 updated for Speijer et al.,**

**2020). While this paper has been in review, another paper has come out to further confirms this through a cyclostratigraphic framework for deposition in the basin (Xu et al., 2023).**

- Line 435: What would be the local factors?

**See our previous comment on influences of local factors.**